# Protective Effect of Treated Olive Mill Wastewater on Target Bacteria and Mitochondrial Voltage-Dependent Anion-Selective Channel 1

**DOI:** 10.3390/antiox12020322

**Published:** 2023-01-30

**Authors:** Paola Foti, Stefano Conti-Nibali, Cinzia L. Randazzo, Simona Reina, Flora V. Romeo, Cinzia Caggia, Vito De Pinto

**Affiliations:** 1Dipartimento di Agricoltura, Alimentazione e Ambiente—Di3A, Università degli Studi di Catania, 95124 Catania, Italy; 2Department of Biomedical and Biotechnological Sciences, University of Catania, 95124 Catania, Italy; 3ProBioEtna srl, Spin Off University of Catania, 95124 Catania, Italy; 4CERNUT, Interdepartmental Research Centre in Nutraceuteuticals and Health Products, University of Catania, 95125 Catania, Italy; 5We.MitoBiotech S.R.L., 95129 Catania, Italy; 6Consiglio per la Ricerca in Agricoltura e l’Analisi dell’Economia Agraria (CREA), Centro di Ricerca Olivicoltura, Frutticoltura e Agrumicoltura, 95024 Acireale, Italy

**Keywords:** olive oil by products, phenols, antioxidant, mitochondria, VDAC, hydrogen peroxide

## Abstract

Olive mill wastewater, a by-product of the olive oil industry, represents an important resource, rich in bioactive compounds with antioxidant activity. In this study, two strategies to concentrate the bioactive components were used: the tangential membrane filtration (ultrafiltration and reverse osmosis) and the selective resin extraction. The concentrates were evaluated for physico-chemical characteristics and antioxidant activity. Furthermore, the antimicrobial activity and the effect on the mitochondrial voltage-dependent anion selective channel 1 were evaluated. The chemical results highlighted that the highest concentration of hydroxytyrosol (as 7204 mg/L) was revealed in the sample obtained by inverse osmosis while the highest concentration of oleuropein (10005 mg/L) was detected in the sample obtained by resin extraction. The latter sample exhibited the highest antimicrobial effects against *Listeria monocytogenes*, *Escherichia coli*, *Staphylococcus aureus* and *Pseudomonas aeruginosa*. Both samples exhibited a high impact on the electrophysiological parameters of VDAC1 activity. These results showed that both valorization techniques, which can be reproduced at industrial scale, provided phenolic concentrates with antioxidant and antimicrobial activity useful for different future perspectives.

## 1. Introduction

Several studies have highlighted the nutraceutical value associated with the Mediterranean diet, with a role in reducing the incidence of chronic degenerative diseases [1]. Among others, one of the key foods in the Mediterranean diet is extra virgin olive oil, a functional food of high nutritional value with a high content of antioxidant molecules [2,3]. However, during the olive oil extraction process only a small portion of the antioxidant content is released in the final product, as most of it is concentrated, during oil extraction, in the resulting by-products. Indeed, only 2% of the initial phenolic compounds present in olives is found in virgin olive oil, while the remaining fraction is found in olive oil by-products, such as about 53% in olive mill wastewater (OMWW) and about 45% in olive pomace (OP) [4]. The discharge of OMWW into the soil or watercourses continues to be a management and economic concern for the Mediterranean countries, due to its phytotoxicity.

In addition to the traditional decantation technique, several purification systems have been proposed as agronomic chemical and biotechnological interventions. Nevertheless, these approaches undervalue ‘waste’ as a potential primary resource of high-value compounds [5]. The phenolic concentrates obtained from olive oil by-products, such as OP, leaves or OMWW, can be used in different application fields, such as food, pharmaceutical and cosmetic industries or in animal feed [3]. In detail, with regard to the food industry, phenols have been proposed as a functional preservative to increase the shelf-life of foodstuffs and improve the health value of final products [6,7,8]. In previous studies, the use of such by-products to isolate standards of bioactive molecules, such as hydroxytyrosol (HT) and tyrosol (TYR), and to reduce toxic substances, such as acrylamide formed during Californian-style processing, has been proposed [9,10]. In particular, OMWW obtained from the three-phase system represents a relevant source of biologically active substances, including HT and TYR. Different strategies have been applied to recover phenolic compounds from OMWW, such as membrane separation [7,11], solvent extraction [12], resins treatment [2,13], centrifugation, chromatographic procedures and enzymatic reactions [14,15]. Regarding the membrane filtration technique, it is one of the most valuable methods, especially at industrial scale, and it is characterised by low energy consumption, good operating conditions and a high efficiency in component separation [16,17]. The extraction in the solid phase represents an interesting technique as it is simple, re-producible and low cost, even if still applied mostly at lab-scale. Both the non-thermal techniques allow the recovery of bioactive compounds preserving their sensory characteristics and nutritional value [18].

Among phenolic compounds, HT is particularly relevant for its antioxidant, anti-inflammatory and antimicrobial activity [19]. In detail, different studies have widely demonstrated that HT acts as a free radical scavenger and metal chelator and presents a key role in protecting against oxidative damage, inhibiting the NADPH oxidase, the inducible form of nitric oxide synthase and the proinflammatory enzymes (5- lipoxygenase and cyclooxygenase), decreasing, in turns, the production of nitric oxide, leukotrienes and prostaglandins [20]. Therefore, OMWW represents a relevant source of phenols to obtain new nutraceutical formulations with antioxidant and antimicrobial activity. The Voltage-Dependent Anion-selective Channel (VDAC) is the main pore-forming protein of the outer mitochondrial membrane (OMM) that, in mammals, exists in three isoforms, numbered as VDAC1, VDAC2 and VDAC3, in the order of their discovery. VDACs play a crucial role in bioenergetics and due to the interaction with Bcl-2-family proteins [21,22,23] and cytosolic proteins, such as hexokinases, glycolytic enzymes, neuronal and cytoskeletal proteins [24,25,26,27,28], they represent the major regulators of mitochondria-mediated cell death and mitochondrial metabolism. Moreover, VDACs differently regulate the homeostasis of Reactive Oxygen Species (ROS) [29,30,31,32]. VDAC1 has been suggested to be essential into ROS-induced apoptosis, as it is responsible for superoxide release from mitochondria. On the other hand, VDAC3 has recently been identified as a mitochondrial ROS sensor that protects the organelle from oxidative stress. Mitochondria are simultaneously the primary source and the main target of ROS; in particular, hydrogen peroxide (H_2_O_2_), a key molecule in the sensing, modulation and signaling of the redox metabolism, depolarises mitochondria and modulates ion channels. Within the well-controlled environment of an artificial membrane, H_2_O_2_ increases the single-channel conductance of VDAC1 [33] although a strong reduction in channel gating within the range of ±50 mV applied has been registered. 

The objective of the present study was to characterise the beneficial effects, as antimicrobial and antioxidant activities, of OMWW samples obtained with different techniques (by tangential membrane filtration and extraction on selective resin) on the main relevant food-related pathogens and on VDAC1. 

## 2. Materials and Methods

### 2.1. Production of Phenolic Fractions from Olive Mill Wastewater

The OMWW was obtained through a three-phase extraction system, from a mix of Nocellara Etnea and Nocellara Messinese cultivars at the Consoli olive oil company (Adrano, Catania, Sicily). The three OMWW fractions, tested in the present study, were obtained by two different methods: tangential membrane filtration and extraction on selective resin (Figure 1). 

#### 2.1.1. Tangential Membrane Filtration System

The samples A and C were obtained by an industrial tangential membrane filtration system. In detail, sample A was obtained by ultrafiltration and sample C was obtained by reverse osmosis. The system used was the ‘Permeaprocess’ (Permeare s.r.l., Milan, Italy) that consists of a tangential filtration based on membranes suitable for both the purification and concentration of compounds. This technique allows the elimination of the water present in the samples (permeate), and the concentration of all the components present.

#### 2.1.2. Extraction on Selective Resin

The OPE sample was obtained, at laboratory scale, by extraction on the selective resin Sepabeads (SP-207) (Mitsubishi Chem. Co., Tokyo, Japan). The resin was treated and loaded into a glass preparative column, according to the method described by Romeo et al. [2]. In detail, the resin was first pre-treated with 95% ethanol (food grade, Carlo Erba, Milan, Italy), then washed with water (HPLC grade, Carlo Erba, Italy) and finally dried at 70 °C until a constant weight was obtained. Then, 20 mL of the dry resin was loaded into a glass preparative column (length, 30 cm; i.d, 0.5 cm) connected to a peristaltic pump (Pump Drive, Heidolph, Schawabach, Germany). The resin was washed down with water, then the OMWW sample was charged until saturation. Before the desorption step, each saturated resin was washed with 80 mL of water to remove water-soluble compounds. The adsorbed phenol was recovered with 40 mL of a 95% ethanol/water solution (60:40, *v*/*v*). Finally, using a rotary evaporator (Rotavapor RE111, Büchi, Cornaredo, Italy), the fraction was concentrated after the vacuum distillation of ethanol at 40 °C.

### 2.2. Chemical Characterization of Samples

The samples, A, C and OPE were subjected to chemical characterization. In particular, pH, total soluble solids (TSS) and total phenols were determined. The pH value was measured using a Mettler DL25 pH meter (Mettler Toledo International Inc., Columbus, OH, USA), while the TSS value, expressed as °Brix, was determined using a refractometer (Atago, RX-5000 Thermo Fischer Scientific, Milan, Italy). The Folin–Ciocalteu’s (FC) colorimetric method was applied to determine the phenolic content. Briefly, samples were mixed with 5 mL of FC reagent (Labochimica, Padova, Italy) and diluted in water at a 1:10 *v*/*v* ratio, with 4 mL of a sodium carbonate solution at 7.5%. After 2 h at room temperature, protected from light, the absorbance of the blue solution was spectrophotometrically measured at 765 nm (Cary 100 Scan UV-Visible, Agilent, Santa Clara, CA, USA). The total phenolic content was expressed as mg gallic acid equivalents (GAE)/L of sample. All analyses were performed in triplicate for each sample.

### 2.3. HPLC Analysis

#### 2.3.1. Phenols

For HPLC analysis, the samples A, C and OPE, were filtered (0.45 µm Millipore filters, Merk Darmstadt, Germany) and injected directly into the HPLC system. The used apparatus was a Waters Alliance 2695 HPLC liquid chromatography with Waters 996 photodiode array (PDA) detector, set at 280 nm and with Waters Empower software (Waters Corporation, Milford, MA, USA). The column, a Luna C18 (250 mm × 4.6 mm i.d., 5 µm, 100 Å; Phenomenex, Torrence, CA, USA) was kept at 40 °C. Chromatographic separation was achieved by gradient elution, using an initial composition of 95% solution A (water acidified with 2% acetic acid) and 5% solution B (methanol) (Merck KGaA, Darmstadt, Germany), at a flow rate of 1 mL/min. A solution of 5 mM of gallic acid (Fluka, Losanna, Switzerland) was used as the internal standard (I.S.). Identification of phenolic compounds was obtained by comparing the retention time with those of pure hydroxytyrosol (HT), tyrosol (TYR) and oleuropein (OLE) standards (Extrasynthese, Genay, France). All analyses were performed in triplicate for each sample.

#### 2.3.2. Organic Acids

Each sample was properly diluted and filtered through a 0.45 μm PTFE syringe filter (Merk, Germany). A total of 5 mM sulphuric acid was eluted in the isocratic mode on a Rezex ROA Organic Acid H+ column (Phenomenex, Torrence, CA, USA). The HPLC instrument (described in the previous section) coupled with a DAD detector was set at 210 nm (with spectrum acquisition in a range of 200 to 400 nm). The run time was set at 50 min at 0.6 mL/min. Lactic, citric, acetic, propionic, isobutyric and butyric acid (Sigma-Aldrich, Milan, Italy) pure standards were injected at different concentrations. All analyses were carried out in triplicate for each sample.

### 2.4. Antioxidant Activity

The samples (A, C and OPE) were diluted and added to a methanolic solution added with 2,2-Diphenyl-1-picrylhydrazyl radical 10^−4^ M (DPPH, Merk, Germany). After 3 min, the absorbance was measured at 517 nm. The results were expressed as a percentage decrease, compared to the control. Inhibition percentage for each sample was calculated as follows:$$\%\;inhibition=\frac{A_{0}-A_{x}}{A_{0}}\;100$$
where *A*_0_ is the absorbance of a DPPH blank, and *A_x_* is the sample absorbance. Antioxidant activity was expressed in relation to the sample volume and the concentration at which 50% of radical scavenging occurred (IC_50_). All analyses were performed in duplicate for each sample.

### 2.5. Antimicrobial Activity

The samples A, C and OPE were tested against the following food-related target strains from ATCC (American Type Culture Collection): *Listeria monocytogenes* ATCC 19114, *Escherichia coli* ATCC 25922, *Candida albicans* ATCC 10231, *Staphylococcus aureus* ATCC 25213, *Pseudomonas aeuroginosa* ATCC 9027, according to Foti and co-workers [7]. Briefly, the tests were performed on Potato Dextrose Agar (PDA, Likson, Palermo, Italy) for *C. albicans* and Muller Hinton Agar Base (MHA, Liofichem, Roseto degli Abruzzi, Italy) for bacteria. The strains, of each individual culture, were standardised using a Mc Farland 0.5 solution for both bacteria and *C. albicans*. Each plate was spatulated with 1 mL of cell suspension and allowed to dry and sterile cellulose discs (Ø 6 mm), imbibed with each tested sample, were placed at different dilution rates. The samples were tested as they were and at different dilution ratios (from 1:2 to 1:8). Distilled water was used as a negative control. Each plate was incubated at specific temperatures for 48 h and then the results were expressed as diameter (mm) of the inhibition halo. The test was performed in duplicate for each strain. 

### 2.6. Electrophysiological Analysis of VDAC Activity

The coding sequence of human VDAC1 (hVDAC1) cloned in the pET21 vector was heterologously expressed in *E. coli* to produce a recombinant His-tagged protein that was subsequently purified using Ni-NTA affinity chromatography (Qiagen, Hilden, DE, Germany) and refolded according to previous studies [27,34,35]. Electrophysiological experiments were carried out at RT in the Planar Lipid Bilayer (PLB) Workstation from Warner Instruments (Hamden, CT, USA) as previously described [36,37,38]. In brief, a solution of 1% of 1,2-diphytanoyl-sn-glycero-3-phosphocholine (DiPhPC Avanti Polar Lipids, Alabaster, AL) dissolved in n-decane was used for bilayer membrane formation on a 200 µm-sized *hole* of a Delrin chamber containing two compartments (i.e., *cis* and *trans*) filled with 3 mL of 1 M KCl, 10 mM HEPES pH 7.4. Bilayer membranes, with an approximately 110–150 pF capacity, were considered satisfactory for subsequent analysis. Reconstitution of hVDAC1 was detected after the addition of ∼40 ng of the purified protein to the *cis* side of the chamber. Channel recordings were performed with the BC-535 Bilayer Clamp amplifier (Warner Instruments) and data were digitalised using the Axon Digidata 1550 Acquisition System (Warner Instruments), with a sampling rate of 10 kHz after low-pass-filter at 300 Hz. Pore conductance (G) was calculated as the ratio of the ionic current through the channel (I) to applied voltage (V) of +10 mV. Voltage-gating parameters of hVDAC1 were investigated by measuring the average conductance of PLB containing a single channel in symmetrical 1 M KCl, 10 mM Hepes, pH 7.4 using a 10 mHz triangular voltage ramp (±50 mV). The conductance (*G*) at a given voltage (from −50 to +50 mV) was normalised to the conductance at the lowest applied potential (−10 mV, *G*_0_). Data are representative of at least three independent experiments and are graphited using prism 8.0 software (GraphPad Software) as the mean ± SEM. Finally, to compare the antioxidant activity the percentage of recovery of VDAC1 voltage-dependence upon the A, C and OPE samples was calculated by acquiring the Area Under Curve (AUC) from G/Gmax plots. Briefly, the AUC values of VDAC1 pre-treated with hydrogen peroxide and subsequently with A, C or OPE were subtracted from the AUC value of VDAC1 pre-treated with H_2_O_2_ and compared to the untreated protein.

### 2.7. VDAC Treatments

The effect of both N-Acetyl-L-cysteine (NAC) (Merck KGaA, Darmstadt, Germany) and tested samples (A, C and OPE) on the membrane alone, was first assessed by adding 8 μM and a final concentration of 1/100 (*v*/*v*), respectively, to both sides of a planar PLB made of 1% Di-Phytanoyl-Phosphatidyl-Choline (DiPhPC). The impact of H_2_O_2_ treatment on single-channel conductance and voltage-dependence was analysed by incubating hVDAC1 with 8 μM of fresh chemical for 30 min at 4 °C. Each tested sample was poured at a final concentration of 1/100 (*v*/*v*) to both sides of a planar lipid bilayer containing peroxide-treated hVDAC1. As a control, NAC was added at a final concentration of 8 μM to both sides of a PLB chamber containing membrane-embedded hVDAC1 pre-treated with H_2_O_2_. Current vs. time traces were recorded in response to constant voltage (+10 mV) and triangular ramp application as described above.

### 2.8. Statistical Analyses

SPSS software (version 21.0, IBM Statistics, Armonk, NY, USA) was used for data processing. One-way analysis of variance (ANOVA) was performed to analyse data and Tukey’s HSD post-hoc test for means separation at a significance level of *p* ≤ 0.05.

## 3. Results

### 3.1. Physico-Chemical Characterisation of Samples

The samples (A, C and OPE) were characterised for their physico-chemical profile: pH, TSS and total phenols (Table 1). The lowest pH values were observed in samples A and C, obtained by the tangential membrane filtration technique, with values of 3.91 and 3.96, respectively. This result is related to the used technology, which concentrates what is present in the matrix, including organic acids. For the same samples (A and C), the TSS value showed the same trend of pH, while the sample OPE exhibited a Brix value lower than that found in sample C. The TSS trend was as expected considering the applied concentration system. The A and C samples, obtained through ultrafiltration and reverse osmosis, respectively, highlighted a total phenol concentration of 3244.11 and 6207.41 mg/L, respectively, whereas the OPE sample showed a total phenol concentration of 16460.42 mg/L. 

### 3.2. Phenols and Organic Acid Detection

Regarding the phenol content, the highest concentration of HT (7203.67 mg/L) was observed in sample C (Appendix A), whereas the highest concentration of OLE was detected (10004.70 mg/L) in OPE sample (Table 2).

Focusing on organic acids, the highest concentration was found in the C sample (Appendix A). Citric and butyric acids were not detected in any samples. Moreover, in the OPE sample, no organic acid should be present, as resin extraction is selective for phenolic compounds. The only acid found was the propionic acid, with a concentration of 1356.50 mg/L (Table 3).

### 3.3. Antimicrobial Activity

The antimicrobial activity of the A, C and OPE samples against target strains was assessed by an evaluation of the inhibition zones (Table 4). Overall, no inhibitory activity was observed against *C. albicans* for any tested sample. Regarding the sample A and C, obtained through ultrafiltration and reverse osmosis, the results exhibited a different antimicrobial activity against pathogens. In detail, while sample A showed an inhibition zone of 7 mm only against *E. coli*, sample C exhibited antimicrobial activity against *E. coli*, *P. aeruginosa* and *L. monocytogenes*. Different behaviors were observed for the OPE sample which exhibited antimicrobial activity against all pathogenic tested strains. 

The concentration of OMWW fractions tested in the presence of halos corresponded to a MIC of 241 mg/mL for the OPE sample against all the tested microorganisms, 507 mg/mL for the C sample against *P. aeruginosa* and *E. coli*, and to 253 mg/mL against *L. monocytogenes*. Sample A was actively undiluted only against *E. coli*. 

### 3.4. Tested Samples Do Not Interfere with Membrane-Reconstituted hVDAC1 under Physiological Conditions

Before evaluating the impact of the tested samples on the electrophysiological properties of hVDAC1, they were added at a final concentration of 1: 100 (*v*/*v*) to both sides of a DiPhPC membrane, generated as previously described. This procedure ensured the exclusion of any undesired disturbance to the phospholipid bilayer, which maintained its stability throughout the analysis time. As shown in Appendix A, the current base line remained near 0 pA upon application of a triangular voltage wave with ±50 mV amplitude.

The addition of the A, C or OPE samples to hVDAC1 inserted into the membrane did not affect the single channel conductance (Appendix A) while it slightly modified the voltage response profile which became noisier (Figure 2C–H), compared to the control (Figure 2A,B). Closure events starting from ±15–20 mV were less clear-cut following administration of the C sample (Figure 2C,D). The A and OPE samples, instead, increased the values of positive voltages required to close the channel (Figure 2E–H). In each case, however, the pore retained its ability to respond to changes in membrane potential.

### 3.5. The Harmful Effect of Hydrogen Peroxide on hVDAC1 Voltage Dependence Is Reversed by A, C and OPE Samples

In order to examine the antioxidant effect of the tested samples, they were added to H_2_O_2_ pre-treated hVDAC1 reconstituted into planar phospholipid membranes at a final concentration of 1: 100 (*v*/*v*). The treatment with hydrogen peroxide had no consequences on the single channel conductance over a long period, although it suppressed the voltage dependence as the current continuously switched between high and low conducting states, without showing any distinct gating event (Figure 3A–F). A, C and OPE samples restored the poregating of hVDAC1 affected by H_2_O_2_: voltages above ±20 mV sharply closed the channel (Figure 3G–M) reestablishing the characteristic ‘bell shaped’ curve of untreated VDAC with lower conductance values at higher membrane potentials (Figure 4A–C).

In particular, the recovery of voltage-dependence induced by C and OPE samples was more pronounced compared to A: the normalised average conductance, G/Gmax, plotted vs. applied voltage demonstrated a complete overlap of the curves from pre-treated hVDAC1 supplemented with C and OPE and the curve from untreated protein (Figure 4B,C*)*. N-Acetyl-Cysteine (NAC), widely known as a powerful antioxidant, was used as a control. After verifying that it did not influence the single channel conductance nor the voltage response of membrane-reconstituted hVDAC1 (Figure 5A), we here report the effect of NAC on hydrogen peroxide-induced loss of voltage dependence which resembled that of tested samples (Figure 5E). Overall, these results suggest that the tested samples possessed antioxidant properties capable of reversing the deleterious effects of H_2_O_2_ on the main functional feature of hVDAC1.

Finally, the antioxidant activity, voltage-dependence recovery and anti-microbial activity of the A, C and OPE extracts were compared. As reported in Table 5 the highest antioxidant capacity of samples C and OPE, detected by DPPH assay (i.e., 41.71 and 50.02 IC_50_ values, respectively), corresponded to an enhanced recovery of the VDAC1 response to the voltage applied (76.27% for C and 78.11% for OPE) and to a broader spectrum of activity against microbial strains.

## 4. Discussion

In this study, two different techniques to recover and concentrate the bioactive compounds from OMWW were applied: the membrane filtration technique (ultrafiltration and reverse osmosis) at industrial scale, according to Foti et al. [7] and the extraction in solid phase, in lab-scale, using adsorbent resins, according to Romeo and co-workers [2]. Based on the results obtained in the present study, the lower pH values were shown by the samples A and C, a result which was highly correlated to the different chemical composition of samples, in particular to the different profile of organic acid content. In fact, while the selective resin extraction technique retained with high affinity the OMWW phenols, the tangential membrane filtration system concentrated phenols together with sugars and organic acids. In particular, a high increase in lactic and acetic acid content was observed in the samples obtained by reverse osmosis. Zooming in on phenolic content, the OMWW sample used to obtain the three tested fractions exhibited a total phenol content of 2163 mg/L (data not shown). Compared to the initial OMWW phenol content, the highest phenol recovery was highlighted by OPE, followed by the C and A samples, with a 4.6, 2.8 and 1.5-fold concentration, respectively. Russo et al. [39], by reverse osmosis, obtained a phenol extract with 8292 mg/L total phenols. Bellumori et al. [40], using reverse osmosis, obtained an extract showing about 12000 mg/L at the same applied temperature. Regarding the HPLC analysis, the highest concentrations were detected in samples C and OPE. In detail, while sample C exhibited the highest content of HT and the absence of OLE, the OPE sample revealed a higher OLE content, reaching concentrations of 10004.7 mg/L, confirming that the concentration of each single phenol is influenced by the applied extraction method. Moreover, the concentration with reverse osmosis, at farm level, occurs over a time longer than that required at lab scale with resins. The longer duration of the extraction process could probably prolong enzymatic (esterase) activity resulting in an increase in HT in sample C. In the present study, the content of HT and TYR detected in sample C (8940 and 1238 mg/L, respectively) was quite similar to that reported by Bellumori et al. [40]. The phenolic compounds have extensive antimicrobial activity, such as antibacterial, anti-viral and anti-fungal effects [41], exerting a beneficial effect on human microbiota [42] and prolonging the shelf-life of food [8]. Several authors have shown that OMWW extracts exhibited an inhibitory activity against both Gram-negative and Gram-positive bacteria, but also on fungi and yeasts [39,43]. In detail, in the present study, a different antimicrobial effect for each sample was revealed, confirming that the antimicrobial mechanism is strongly related to the phytocomplex present in the different matrices. In fact, while samples A and C were characterised by all molecules present in the starting matrix (such as phenols and organic acids), the sample OPE, obtained by adsorption technology, was mostly composed of phenolic fraction. In the present study, the samples C and OPE exhibited greater inhibitory activity than sample A, which displayed modest inhibitory activity exclusively against *E. coli*. As matter of fact, both samples C and OPE showed a wider spectrum of action, with sample C able to inhibit *E. coli*, *P. aeuroginosa* and *L. monocytogenes*, and the sample OPE able to inhibit *S. aureus*. These results, related to the different composition of the two samples, appear strongly related to the different OLE content. Indeed, several studies have shown the antimicrobial effect of OLE as higher against Gram-positive than Gram-negative bacteria because of the difference in cell structures [44,45]. Topuz and co-workers [46] showed that *S. aureus* was sensitive to different OLE extracts at the different tested concentrations (3.125 mg/mL and 0.781 mg/mL). Furthermore, in the present study, all samples showed no antagonistic effect against *C. albicans*. This result could be due to the hydrophilic nature of the tested samples according to previous reports that have shown that the alteration of the hydrophilic/lipophilic balance could affect the cellular uptake mechanisms, enhancing antioxidant or antimicrobial activities [47]. Despite the high OLE content in the sample OPE, the highest antioxidant activity was observed in sample C. This result is probably related to the higher content of HT. Several reports have praised the efficiency of HT and OLE in reducing the production of free radicals [48,49,50,51]. In particular, OLE is also known to cause mitochondria-mediated apoptosis in cancer cells [52,53,54]. Although many reports have emphasised the ability of HT and OLE to reduce oxidative stress and to improve mitochondrial biogenesis and function, no data regarding the effects of such antioxidants on mitochondrial proteins are still available. Therefore, samples A, C and OPE were tested to assess their antioxidant effect on hydrogen peroxide-treated human VDAC1, reconstituted into a planar phospholipid membrane. As previously described, VDAC is one of the most relevant proteins of the outer mitochondrial membrane that has been demonstrated to play a key role in apoptosis and ROS homeostasis. H_2_O_2_, in turn, is mainly produced within mitochondria [54], which also represent the main target of its harmful effects [55]. According to Malik and Ghosh [33], the results of the present study revealed a complete loss of VDAC1 voltage-dependence upon H_2_O_2_-induced overoxidation. The tested samples proved to be able to antagonise and reduce the negative outcomes of hydrogen peroxide treatment on the electrophysiological features of VDAC1. In this regard, samples C and OPE, most likely due to the higher phenol content, yielded the best results in terms of recovery of the channel’s ability to respond to changes in voltage applied. They completely re-shaped the normalised average conductance plotted vs. applied voltage of H_2_O_2_-treated VDAC1, making it indistinguishable from the curve of the untreated protein. Those results perfectly resembled data obtained with N-acetyl-cysteine whose ability to minimise oxidative stress is largely known. Consequently, the recovery of the oxidation of VDAC, the gateway to the mitochondrion for all substrates and metabolites, predicts a positive effect on the organelle’s respiratory and bioenergetic function.

## 5. Conclusions

The effect of two methods, such as tangential membrane filtration and extraction with absorbent resin, in obtaining a matrix rich in bioactive compounds was evaluated. The results showed that both techniques (reverse osmosis and resin extraction) had a positive effect on the concentration of bioactive components, and although a different antimicrobial activity, related to the different phenolic profiles, was observed, no difference in antioxidant activity was detected. These results are relevant to drive the choice of the most suitable technique to treat OMWW, taking into account the criteria of cost-effectiveness and productivity. Both extraction techniques, indeed, involve several production costs related to the purchase of the necessary plants and, in the case of resin extraction, also to the use of organic solvents (food-grade ethanol). Moreover, the results of the present study suggest that further effort must be made to better explore the antioxidant mechanism of bioactive compounds present in olive oil by-products in order to propose natural food preservatives or food supplements with nutraceutical properties.

## Figures and Tables

**Figure 1 antioxidants-12-00322-f001:**
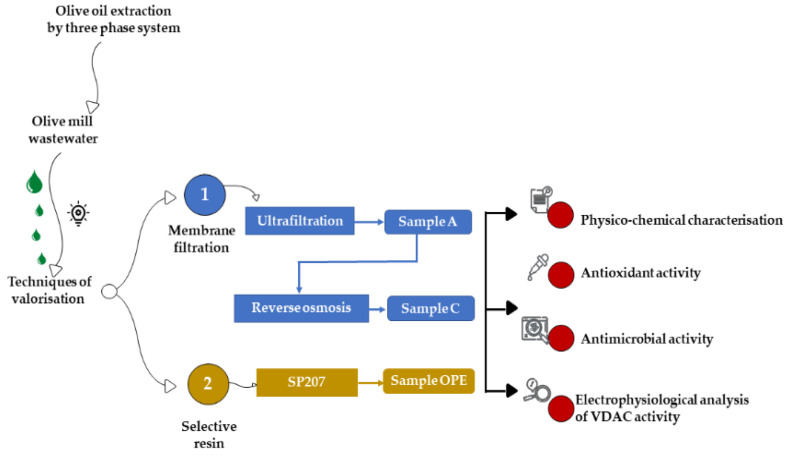
Experimental design.

**Figure 2 antioxidants-12-00322-f002:**
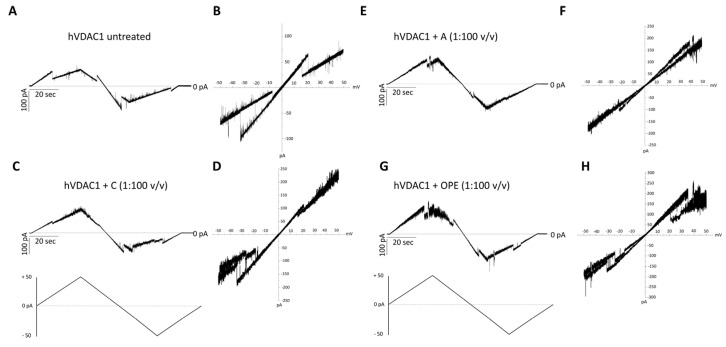
Analysis of hVDAC1 voltage dependence following the addition of the tested com-pounds by triangular voltage ramps. Sample A (panels (**E**,**F**)), sample C (panels (**C**,**D**)) and sample OPE (panels (**G**,**H**)) were added to both sides of a DiPhPC bilayer containing hVDAC1 at a final concentration of 1:100 (*v*/*v*) and compare with the control (panels (**A**,**B**)). Current measurements were performed in symmetrical 1 M KCl upon application of a triangular voltage ramp of ±50 mV. The corresponding I–V plots were obtained by plotting the current as a function of clamp voltage.

**Figure 3 antioxidants-12-00322-f003:**
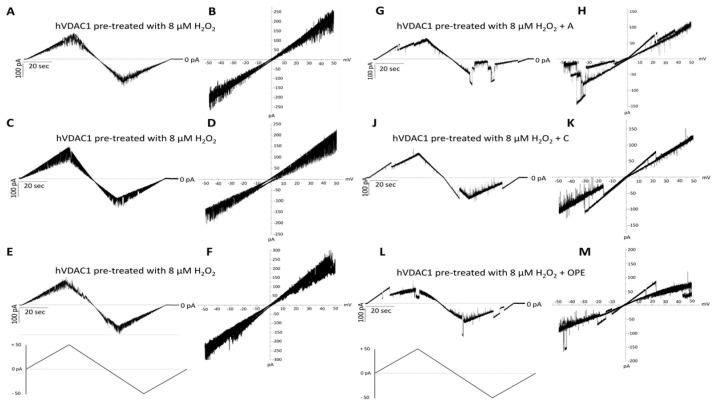
Analysis of the effect of tested samples on hVDAC1 voltage dependence following hydrogen peroxide treatment by triangular voltage ramps. Sample A (panels (**G**,**H**)), sample C (panels (**J**,**K**)) and sample OPE (panels (**L**,**M**)) were added to both sides of a DiphPC bilayer containing hVDAC1 pre-treated with 8 µM H_2_O_2_ (panels (**A**–**F**)) at a final concentration of 1:100 (*v*/*v*). Current measurements were performed in symmetrical 1 M KCl upon application of a triangular voltage ramp of ±50 mV. The corresponding I–V plots were obtained by plotting the current as a function of clamp voltage.

**Figure 4 antioxidants-12-00322-f004:**
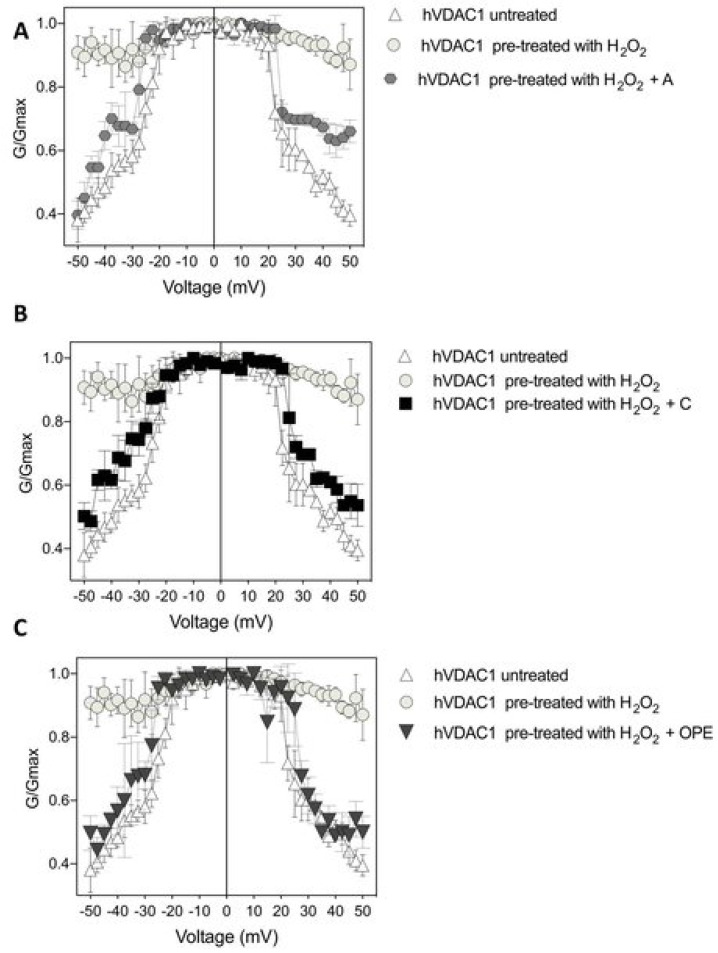
Conductance G/Gmax of hVDAC1 untreated (**A**–**C**), pre-treated with 8 µM H_2_O_2_ (**A**–**C**) and challenged with sample A (panel (**A**)), sample C (panel (**B**)) and sample OPE (panel (**C**)) as function of the applied voltage. Data are expressed as mean of at least 4 independent experiments.

**Figure 5 antioxidants-12-00322-f005:**
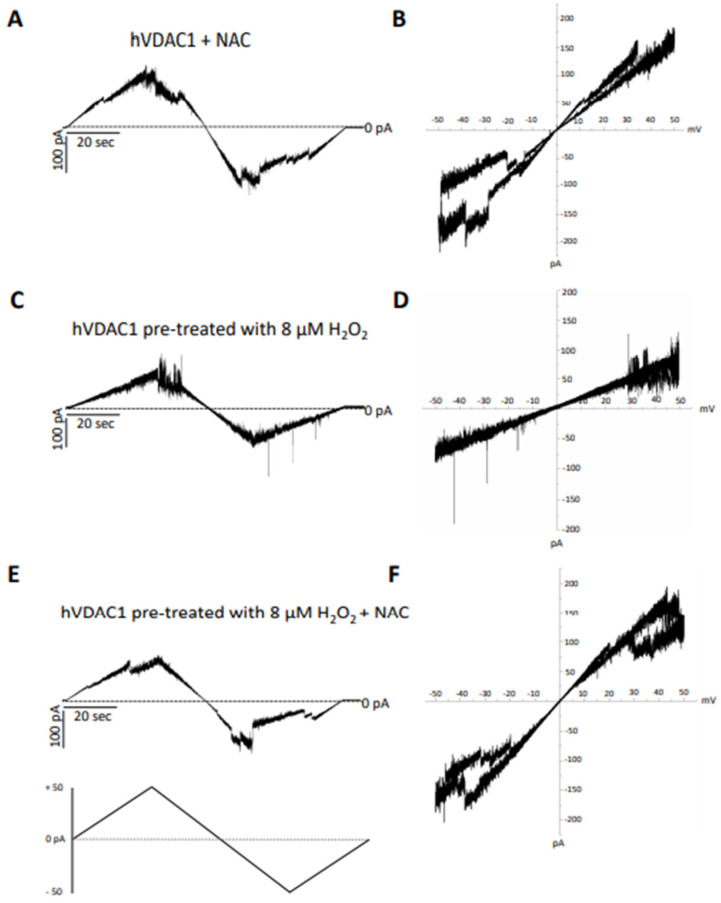
Analysis of the effect of N-acetyl cysteine on hVDAC1 voltage dependence following hydrogen peroxide treatment by triangular voltage ramps. NAC was added at a final concentration of 8 µM to membrane-embedded hVDAC1pA pretreated (panels (**C**–**F**)) or not (panels (**A**,**B**)) with 8 µM H_2_O_2_ (panels (**C**,**D**)). Current measurements were performed on symmetrical hVDAC1 pre-treated with 8 µM H_2_O_2_ 1 M KCl upon application of a triangular voltage ramp of ±50 mV. The corresponding I–V plots were obtained by plotting the current as a function of clamp voltage.

**Table 1 antioxidants-12-00322-t001:** Chemical characterisation of the three tested OMWW fractions.

Sample	pH	TSS (°Brix)	Total Phenols (mg/L)
A	3.91 ± 0.03 ^c^	6.29 ± 0.05 ^c^	3244.11 ± 0.21 ^c^
C	3.96 ± 0.06 ^b^	10.63 ±0.04 ^a^	6207.41 ± 0.11 ^b^
OPE	4.80 ± 0.06 ^a^	7.94 ± 0.03 ^b^	16460.42 ± 11.3 ^a^
	**	**	**

Data are expressed as means ± SD. Mean values with different letters within the same column are statistically different. ** Significance at *p* ≤ 0.01. Legenda: Sample A = obtained by ultrafiltration; sample C = obtained by reverse osmosis; sample OPE = obtained by resin extraction.

**Table 2 antioxidants-12-00322-t002:** Phenol detected in the three tested OMWW fractions.

Sample	HT (mg/L)	TYR (mg/L)	OLE (mg/L)
A	3414.96 ± 0.21 ^c^	494.37 ± 0.12 ^c^	0.00 ± 0.00 ^b^
C	7203.67 ± 0.31 ^a^	1046.62 ± 0.24 ^b^	0.00 ± 0.00 ^b^
OPE	3240.50 ± 0.25 ^b^	2015.54 ± 0.31 ^a^	10004.70 ±0.02 ^a^
	**	**	**

Data are expressed as means ± SD. Mean values with different letters within the same column are statistically different. ** Significance at *p* ≤ 0.01.

**Table 3 antioxidants-12-00322-t003:** Results of organic acids of the three tested OMWW fractions analysed by HPLC.

Samples	Lactic Acid (mg/L)	Acetic Acid (mg/L)	Propionic Acid (mg/L)	Isobutyric Acid (mg/L)
A	3554.3 ± 58.78 ^b^	3554.3 ± 58.78 ^b^	0.00 ± 0.00 ^c^	12621.7 ± 374.88 ^a^
C	7953.7± 7.93 ^a^	12137.2 ± 7.38 ^a^	2984.4 ± 89.77 ^a^	0.00 ± 0.00 ^b^
OPE	0.00 ± 0.00 ^c^	0.00 ± 0.00 ^c^	1356.50 ± 87.0 ^b^	0.00± 0.00 ^b^
	**	**	*	**

Data are expressed as means ± SD. Mean values with different letters within the same column are statistically different. * Significance at *p* ≤ 0.05; ** Significance at *p* ≤ 0.01.

**Table 4 antioxidants-12-00322-t004:** Antimicrobial activity (expressed as halo diameter of inhibition in mm).

Target Strains	Sample	Sample Dilution			
		Raw	1:2	1:4	1:8
*Escherichia coli*	A	7	-	-	-
ATCC 25922	C	8	7	-	-
	OPE	11	9	7	-
*Pseudomonas aeruginosa*	A	-	-	-	-
ATCC 9027	C	9	7	-	-
	OPE	10	7	7	-
*Staphylococcus aureus*	A	-	-	-	-
ATCC 25213	C	-	-	-	-
	OPE	8	7	7	-
*Listeria monocytogenes*	A	-	-	-	-
ATCC 19114	C	8	7	7	-
	OPE	10	8	8	-

**Table 5 antioxidants-12-00322-t005:** Comparison of the antioxidant activity, voltage-dependence recovery and anti-microbial activity of each tested sample.

Sample	Antioxidant Activity (IC_50_)	Recovery of VDAC1 Vdep	Anti-Microbial Activity
A	84.00 ± 0.12	62.20 ± 0.45%	*Escherichia coli* ATCC 25922
C	41.71 ± 0.03	76.27 ± 0.47%	*Escherichia coli* ATCC 25922 *Pseudomonas aeruginosa* ATCC 9027 *Listeria monocytogenes* ATCC 19114
OPE	50.02 ± 0.04	78.11 ± 0.72%	*Escherichia coli* ATCC 25922 *Pseudomonas aeruginosa* ATCC 9027 *Listeria monocytogenes* ATCC 19114 *Staphylococcus aureus* ATCC 25213

## Data Availability

All related data and methods are presented in this paper. Additional inquiries should be addressed to the corresponding author.

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
