# Peer review of "Protective Effect of Treated Olive Mill Wastewater on Target Bacteria and Mitochondrial Voltage-Dependent Anion-Selective Channel 1"

_antioxidants, 2023, doi:10.3390/antiox12020322_

Round 1
Reviewer 1 Report
The manuscript entitled "Protective effect of treated olive mill wastewater on target bacteria and mitochondrial Voltage-Dependent Anion-selective Channel 1" (Foti et al) report the chemical and biological characterization of three OMWW extracts (namely A, C and OPE). The extracts were obtained through different methods (i.e., ultrafiltration, reverse osmosis and resin extraction) and showed different antioxidant and antibacterial properties.
The following observations have been raised during the review:
Lines 39-41. This statements should be supported by proper references. Moreover, the environmental impact of OMWWs should be mentioned in the introduction
Please add a figure with the chemical structures of HT, TYR and OLE
Section 2.1. Details on the cultivar, region growing conditions of the olives are requested. Additionally, more details on the extraction processes are needed.
Section 2.3. The HPLC chromatograms for phenol and organic acids identification should be reported either in the manuscript or in the supplementary material
Line 149. DiPhPC acronym is not defined.
Table 2. For sake of clarity, please report the antioxidant activity percent (AA%) of the three analyzed fractions.
Table 3. The meaning of a single star “*” (4th column, last row) is not defined
Section 3.3. The antimicrobial activity has been reported as dimension (mm) of the inhibition zones. For sake of clarity, please express the activity as MIC value. Furthermore, the reported activities do not appear to be dose-related. The authors should comment on this
Section 3.3. More details on the biological meaning of VDAC1 testing are needed. The authors demonstrated that fractions C and OPE have antioxidant properties being able to counteract the effects of H2O2 on the considered channels. However, this piece of information was already collected by the DHPP assay.
Lines 313-336. The authors correlate the antibacterial properties of the fraction to the OLE content. This appear to be quite a gross consideration given the complexity of the analysed mixtures. Furthermore, the antioxidant and antibacterial properties of OMWW extracts have been already reported in the literature (e.g. Russo et al., Antioxidants, 2022, 11, 903). The authors should highlight the novelty of their findings and comment their results on the basis of the data already available to the scientific community.
Line 342-346. A quantitative correlation between the VDAC oxidation and DPPH values is missing.
Lines 380-389. Please correct
Author Response
Please, see the attachment

Reviewer 2 Report
The manuscript “Protective effect of treated olive mill wastewater on target bacteria and mitochondrial Voltage-Dependent Anion-selective Channel 1" is devoted to the study two strategies to concentrate the bioactive component of the olive mill wastewater. Tangential membrane filtration (ultrafiltration and reverse osmosis) and selective resin extraction had been studied. The bioactive compounds obtained were evaluated for their physico-chemical parameters and their antioxidant activity.
In general, the manuscript makes a good impression, but a lot of questions should be resolved to improve the quality of the manuscript. More details are not described enough. Thus, some revisions should be done to improve the quality of the manuscript.
· Abstract. Lines 30-31. You have to delete or readapt this missing paragraph.
Lines 25-26. This amount of phenols are high, medium or less level compared with oil, table olives, olive leaf extracts?
I would include a general conclusion at industrial level of the results obtained.
· Introduction: A deep description of the antecedent of the work has been displayed in the manuscript. That background helps significantly to understand the rest of the article.
I would include a large paragraph related to the ways to revalue the bioactive compounds in this industry by applying it in other foods. In this paragraph, I would include ways to ways to revalue the by-products of this agro-industry and also with olive mill wastewater. You can find a lot of literature in which describes the application of phenolic extracts obtained from the olive growing industry and their revaluation in agro-industrial foods. You can isolate standards of HTy or Ty (Martín-Vertedor et al., 2020. Food Control), you can add the extracts obtained from the olive industries in different foods to improve the amount of phenols (Baccouri et al. 2022. Food Analytical Method) or even to reduce toxic substances (Mechi et al. 2023. Antioxidants). It is important to highlight the importance of the practical application of the work that you are proposing on an industrial scale. Is your work useful for the industry?
· Materials and Methods: What kind of variety was used?
How many repetitions did you do?
2.1 epigraph. You should explain with more detail each of the techniques used. I would include some sub-epigraph.
Which was the experimental design?
At what wavelength were the studied phenolic compounds detected?
You should indicate the name of all the equipment used: brand, city, country, etc.
You should indicate the same with the microbiological strains used in this study.
· Results and Discussion: This part must be reviewed in depth by the authors.
Table 1. I should include the description of A, C and OPE samples in the bottom of the table.
The description of 3.1 epigraph is too poor. What about the pH or ºBrix even significant differences occurs?
I would change mg/L by mg∙L-1.
Table 3. more description of the content of the table should be necessary. Line 205.
The sample with highest total phenol has not highest HT, TYR or OLE? Why?
The sample with highest phenol has more antimicrobiological activity?
You should discuss the results obtained in your assays with those describe in the bibliography. The amount of phenols are high or less compared to other olives extracts or foods?
Lines 265-268. Remove italics.
Lines 284-288. It seems like introduction part. I would explain the general results and after that I will discuss with this information relating your data with those previously described in the bibliography.
Lines 289-292. It seems like material and methods. I think that this part has been described in material and method section.
Lines 293-300. It seems like introduction part.
I like the fourth paragraph. I would discuss the results presented in this paragraph with those explain the previous paragraph.
Line 301-310. Why sample C has highest content in HT? You should explain the results with the bibliography.
Line 311-313. I would start the paragraph with lines 313-314. After that, I would justify the effect of phenol, Lines 301-310.
Why some samples have antimicrobiological activity for one or other microorganism?
Why do not measure sensory analysis? I think that it could be interesting for future works. In this way you can think about the use of this extract in the agroindustry.
Do you think these improvements could be implemented at an industrial level?
· Conclusions: It could include some of the disadvantages of using these technique for future studies. I would like you to describe properly the method of extraction of these phenols in material and method. I do not know if you use organic solvents? In this case, you have to take it into account in the conclusion part.
I like this part of the conclusion: Lines 359-362. It is in line with the paragraph that I asked you to describe in depth in the introduction part.
Lines 380-389. Please review this part. You have copied the information of the instruction of the Journal.
Author Response
Please, see the attachment.

Round 2
Reviewer 1 Report
The authors amended the manuscript according to the provided suggestions and improved the quality of their work. Still, some issues need to be addressed before publication as detailed blow:
Line 73: add full stop after [9-10]
Line 87: please replace “the HT” with “HT”
Line 171: please add a blank space after TSS
Supporting information: the chromatograms of phenols and organic acids of all analysed samples (namely, A, C and OPE) should be included. Additionally, the supplementary file should be organized as a single doc/PDF file and a caption for each figure should be included.
Line 388: please replace “as is (undiluted)” with “undiluted”
Table 5 and lines 388-395: please include the table and its comment in the “Discussion” section. In the current position it generates confusion as anticipate data (namely, VDAC recovery and DPPH results) not yet reported in the manuscript.
Line 502: please correct “Compared t to”
Line 555: please replace “Hence, after evaluating physico-chemical properties and antimicrobial activity” with “Despite many reports have emphasized the ability of HT and OLE to reduce oxidative stress and to improve mitochondrial biogenesis and function, no data regarding the effects of such antioxidants on mitochondrial proteins are still available. Therefore, samples ….”
Author Response
Please, see the attachment

Reviewer 2 Report
The authors have made a number of changes to the text that significantly improve the quality of the publication. The language used in the article has been improved. Thoughts and facts were exposed clearly. The well-written manuscript about has a significant relevance to food science research field. The number of current bibliographical references has been increased. Therefore, I believe that all necessary corrections have been made by the authors and the article should be considered for publication in the Journal. I have no comments on the current content of the article.
Author Response
Thank you very much for your comments.